# The atmospheric settling of commercially sold microplastics

Alina Sylvia Waltraud Reininger[1], Daria Tatsii[1], Taraprasad Bhowmick[2], Gholamhossein Bagheri[2], and Andreas Stohl[1]

[1]Department of Meteorology and Geophysics, University of Vienna, Josef-Holaubek-Platz 2, 1090 Vienna, Austria
[2]Laboratory for Fluid Physics, Pattern Formation and Biocomplexity, Max Planck Institute for Dynamics and Self-Organization, Am Fassberg 17, 37077 Göttingen, Germany

**Correspondence:** Alina Sylvia Waltraud Reininger (alina.reininger@univie.ac.at)

**Abstract.** The atmosphere plays a major role in the dispersion of microplastics in the environment. The atmospheric transport of large microplastics is strongly influenced by their settling behavior, which depends on their physical properties, including size and shape. However, experimental data on the settling behavior of commercially available microplastics with complex, nonspherical shapes in air are rare. Here we present experiments on the gravitational settling velocity of commercially available glitters (nominal diameters between 0.1 and 3 mm) and fibers (lengths between 1.2 and 5 mm). We observed that glitters and fibers settle up to 74 % and 78 % slower compared to volume-equivalent spheres, respectively. The atmospheric transport of fibers has been studied previously; however, there are no studies on the atmospheric transport potential of glitters. Therefore, we used an atmospheric transport model constrained by our experimental results to assess the transport potential of glitters. Our results reveal that glitters exhibit transport distances 12–261 % greater than volume-equivalent spheres, highlighting their elevated atmospheric transport potential. As a result, the environmental impact of glitter particles, especially following their use in entertainment events, warrants attention and mitigation.

## 1 Introduction

Microplastics, synthetic polymers ranging from 1 to 5000 µm in size, are a major source of pollution worldwide (Petersen and Hubbart, 2021). They can be categorized as either primary microplastics - manufactured intentionally for products such as cosmetics, cleaning agents, and personal care products (Auta et al., 2017) - or secondary microplastics, formed by the fragmentation of larger plastics (Petersen and Hubbart, 2021). Shapes of microplastics include films, irregular particles, fibers, spheres, and spheroids (Hartmann et al., 2019), with the observed shapes often differing by location (Zhang et al., 2020; Petersen and Hubbart, 2021). In many studies of urban locations, for example, fibers have been identified as the most common shape of microplastics (Dris et al., 2015; Liu et al., 2019a; Zhou et al., 2017; Cai et al., 2017). Fibers are also the dominant shape observed in several remote locations (Brahney et al., 2020; Cabrera et al., 2020; Ambrosini et al., 2019). Similar to fibers, films have been detected in both urban (Abbasi et al., 2019; Cai et al., 2017; Zhou et al., 2017) and remote locations (Allen et al., 2020; Cabrera et al., 2020). Glitter - small, flat, reflective particles ranging from 50 up to 6350 µm, with 200 µm being the most popular size (Yurtsever, 2019b; Blackledge and Jones Jr., 2007) - is classified as a primary microplastic. Glitter particles can be described as films or irregular particles according to the nomenclature recommendations of Hartmann et al.

(2019), or disks, as done by Tinklenberg et al. (2023, 2024). The production typically involves plastics, metal foils, or dyes (Tagg and Ivar do Sul, 2019; Zanini et al., 2024). As an alternative, renewable, plant-based raw material, such as seaweed, regenerated cellulose, mica powders, and glycerin are produced (Yurtsever, 2019b; Zanini et al., 2024). According to the report of Market Research Intellect (2025), the glitter powder market was valued at USD 100 Million in 2023 and is expected to reach USD 324.34 Million by 2031. The growth in glitter powder industry can be attributed to its broad application and an

increasing demand in many sectors, such as the automobile, textile, and cosmetic industries (Market Research Intellect, 2025; Zanini et al., 2024). Furthermore, there is a growing interest in crafting activities where glitter powder is added to decorations, gift cards, and arts (Market Research Intellect, 2025; Yurtsever, 2019a). Glitter is also extensively used in the entertainment sector for decorations, stage sets, special effects, and events such as carnivals and shows (Yurtsever, 2019a; Zanini et al., 2024; Market Research Intellect, 2025).

Microplastics are associated with substantial harmful effects on the environment. Several studies have investigated the adverse effects (e.g., acute toxicity, increased susceptibility to pollutants and mortality, reduced food intake, and potential bioaccumulation) of microplastic fibers on soil and aquatic organisms (Gray and Weinstein, 2017; Romanó de Orte et al., 2019; Kwak et al., 2022; Song et al., 2019; Kim et al., 2021). Furthermore, airborne fibers have the potential to harm the human respiratory systems (Prata, 2018). However, glitter has received less scientific attention than fibers (Yurtsever, 2019b; Zanini

et al., 2024). Glitter has been reported in riverbed and coastal sediments, surface waters (Nithin et al., 2022; Ballent et al., 2016), wastewater treatment plants (Lusher et al., 2017; Raju et al., 2020), and street dust (Abbasi et al., 2017, 2019; Dehghani et al., 2017). Due to the recycling of biosolids, glitter poses a threat to farm soil and soil invertebrates (Harley-Nyang et al., 2022; Dąbrowska, 2022; Medyńska-Juraszek and Jadhav, 2022; Chen et al., 2024). Both conventional and biodegradable glitter cause ecological impacts in aquatic ecosystems (Green et al., 2021; Luana Lume Yoshida and da Cunha-Santino, 2023). In

aquatic organisms, glitter is associated with internal damage, adverse effects on embryonic development, biovolume increment, an increase in antioxidant defense, and reducing survival, feeding, hatch, and growth rates (Abessa et al., 2023; Machado et al., 2023; Provenza et al., 2022; Das Pramanik et al., 2023).

For transporting microplastics to different environmental media, the atmosphere plays an important role (Dris et al., 2016; Allen et al., 2020; Bergmann et al., 2019; Liu et al., 2019b; Trainic et al., 2020; Wright et al., 2020; Huang et al., 2021;

Mandal et al., 2024; Martynova et al., 2024). Particle shape is a key factor influencing the atmospheric transport potential of microplastics, primarily through its effect on terminal settling velocity $v_t$. A particle obtains its terminal settling velocity when the force of gravity is balanced by the drag force. $v_t$ is calculated by:

$$v_t = \left[ \frac{4}{3} C_{cun} g \frac{d_{eq}}{C_D} \frac{\rho_p}{\rho_a} \right]^{1/2}, \tag{1}$$

where $g$ is the gravitational acceleration, $C_{cun}$ is the Cunningham slip correction factor (which accounts for reduced drag on

particles < 1 μm), $d_{eq}$ is the volume-equivalent diameter of a sphere, $\rho_p$ and $\rho_a$ are the densities of the particle and the air, respectively, and $C_D$ is the dimensionless drag coefficient, which depends on the particle's shape and orientation (Bagheri and Bonadonna, 2016).

Only a limited number of experimental studies have examined the settling behavior of microplastic films, disks, fibers, and fragments in air (Qi et al., 2012; Preston et al., 2023; Tatsii et al., 2024; Tinklenberg et al., 2023, 2024; Musso et al., 2024). This study is the first to investigate the atmospheric transport potential of glitter in the size range of 50 µm to 3 mm. For that, we conducted laboratory experiments to measure the gravitational settling behavior of microplastic glitters of six sizes. Additionally, we examined the settling behavior of eight sizes of commercially available microplastic fibers with lengths ranging from 1.2 to 5 mm. Such fibers, in contrast to the 3D-printed fibers used by Tatsii et al. (2024), can contain imperfections that might impact their settling behavior. These experimental findings were then integrated into an atmospheric transport model to evaluate the residence times and travel distances of glitter particles, as the atmospheric transport potential of fibers has been investigated in detail by Tatsii et al. (2024).

## 2 Methods

### 2.1 Particle properties

Experiments were performed for six different sizes of glitter particles (craft glitter, Hemway), characterized by hexagonal or irregular shapes. These glitter particles have nominal diameters of 0.1, 0.2, 0.4, 0.6, 1, and 3 mm, with longest dimensions $L$ from 0.18 - 3.29 mm (Table 1). Their densities were determined with a helium pycnometer (AccuPyc II 1340 V3.00, Micromeritics Instrument Corporation), resulting in an average density of 1.39 g/cm$^3$, ranging from 1.38 - 1.40 g/cm$^3$. The glitter particles' dimensions were determined using a 3D laser microscope (VK-X260K, KEYENCE International (Belgium) NV/SA) and the software ImageJ (Schneider et al., 2012). The sphericity $\Psi$, which compares the surface area of a sphere to that of a nonspherical particle with equivalent volume, ranges from 0.83 to 0.19. It is defined as $\Psi = \pi d_{eq}^2 / SA$, with $SA$ being the surface area of the particle.

The fibers consist of Polyamide 6.6 Precision Cut Flock (Flockan) with a density of 1.15 g/cm$^3$. Eight fiber sizes were studied, with lengths ranging from 1.2 to 5 mm and diameters between 10 and 105 µm (Table 2). The fibers possess aspect ratios AR (longest dimension $L$ / shortest dimension $S$) between 36 and 121 and no or a small curvature. The sphericity ranges from 0.11 to 0.01. Microscopic images of the glitter particles and fibers are presented in Fig. 2(a)-(d) and Fig. S1-S3.

### 2.2 Experimental setup

In our experimental setup, one can measure the three-dimensional translational and rotational motion of solid particles between 0.1 - 5 mm in quiescent air. The setup consists of an air-filled settling chamber, where the particles can settle under gravity, surrounded by four high-speed cameras. The settling chamber is a reinforced steel chamber with four glass windows, airtightly sealed edges, and dimensions of $90 \cdot 90 \cdot 200$ mm in the X (direction of the light path from the LED), Y (horizontal direction orthogonal to X), and Z (direction of gravity) directions. A detachable bottom drawer allows for collection of the particles after an experiment. A particle injector is mounted on the settling chamber's removable top cover. The settling chamber is

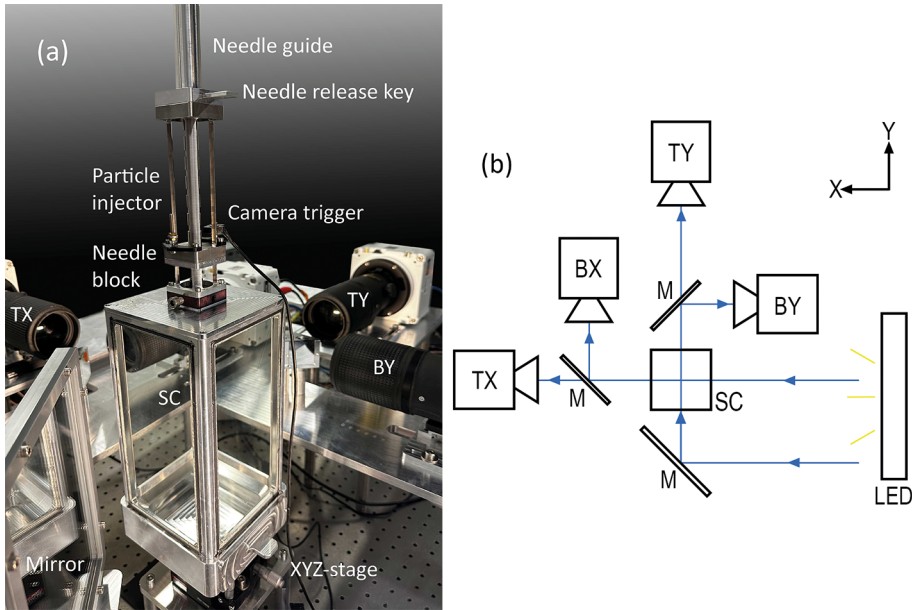

**Figure 1.** Experimental setup. (a) Photo of the settling chamber (SC) with the particle injector and the XYZ-stage. Three cameras (TX, TY, and BY) and a mirror can be seen. (b) Schematic view of the experimental setup showing the two upper cameras (TX) and (TY), the two lower cameras (BX) and (BY), the mirror (M) arrangement, as well as the settling chamber (SC), the LED, and the illumination/imaging paths.

installed on a high-precision XYZ-stage, which allows movement of the settling chamber with $10\,\mu m$ spatial resolution in all three directions Bhowmick et al. (2024a). A photograph of the settling chamber is shown in Fig. 1 (a).

The four high-speed cameras (Phantom VEO4K 990L, Vision Research) synchronized with a high-frequency pulsed white LED array (LED-Flashlight 300, model number 1103445, LaVision GmbH) surround the settling chamber, as shown in Fig. 1 (b). Only one camera receives direct light from the LED, the remaining three receive light reflected by mirrors. A waveform generator controls the pulse rate, amplitude, offset of the waveform, and duty cycle of the LED and creates a synchronization signal for the exposure times of all cameras. To control the cameras, the Phantom camera control (PCC) software was used.

Further details of the setup, as well as the postprocessing and calibration processes, are described in Bhowmick et al. (2024a) and Tatsii et al. (2024). For the experiments, the resolution of the cameras was set to $4096 \times 1140$ pixels, each pixel corresponding to a physical area of $6.75 \times 6.75\ \mu m^2$, however, the smallest dimensions of the tested particles are larger than this value. The exposure time was set to $5\,\mu s$. For samples 1-4 of the fibers, a frame rate of 200 frames per second was chosen. For all other particles, the frame rate was set to 1492 frames per second.

Prior to the experiments, the model of Bagheri and Bonadonna (2016, 2019) was used to estimate the vertical distance at which the particles approximately reach their terminal velocity. The settling chamber was positioned accordingly using the XYZ-stage to allow the cameras to capture particles near their steady-state velocity. After calibration, particles were introduced into the settling chamber using the particle injector. For this, a single particle was placed on top of a particle injector needle

(a cylindrical rod with a length of $200\,\mathrm{mm}$, a diameter of $12\,\mathrm{mm}$, and a conical tip Bhowmick et al. (2024a)), which was then inserted into a needle guide. A release key that was placed into one of the needle grooves secured the needle in place. Upon removal of the key, the needle dropped vertically until stopped by a needle block, causing the particle to detach. Different grooves represent different particle initial velocities, ranging between $0.42 - 1.5\,\mathrm{m/s}$ Bhowmick et al. (2024a). Notice that insertion speeds are chosen to be close to expected terminal settling velocities, so particles can accelerate or decelerate in the air column, depending on whether insertion is slower or faster than the terminal settling velocity. As soon as the particle detached, cameras were activated by an external photoelectric trigger. When a particle was observed by all four cameras, the images were stored for postprocessing. Figure 2(e)-(h) and Fig. S4-S5 illustrate overlays of particle images recorded by the cameras.

## 2.3   Verification of the setup

The sensitivity tests of the experimental setup by Bhowmick et al. (2024a) showed that possible uncertainties in measurements caused by a change of temperature due to illumination and airflow caused by needle movement are negligible. Bhowmick et al. (2024a) and Tatsii et al. (2024) both thoroughly verified the setup by dropping spheres and nonspherical particles of different diameters and comparing the measured settling velocities to the empirical model of Clift and Gauvin (1971) and to the direct numerical simulations of (Bhowmick et al., 2025). The authors report relative errors below 5%. We point out that the stated error of the Clift and Gauvin (1971) model is about 6 % at Reynold's numbers below $3 \cdot 10^5$ (Clift and Gauvin, 1971) and the average error of the Bagheri and Bonadonna (2016) model is 10 %.

## 2.4   Postprocessing and data analysis

Following the method outlined in Bhowmick et al. (2024a), for each camera, a median image background was calculated using all available frames and subsequently subtracted from the images. Then, a small region around the particle was cropped, and the resulting image was independently thresholded. The centroid position and longest length were extracted from the thresholded image using the OpenCV 2 library (Open Source Computer Vision Library) in Python. The centroids were used to determine the three-dimensional positions of the particles, and the velocities were extracted using a sixth-order finite difference scheme.

To ensure that the terminal settling velocity had been reached, only experimental runs with relative changes in velocity of less than 10 % over a wide number of frames were chosen. Figure 2(i)-(l) shows the postprocessed settling velocities corresponding to the particles shown in Fig. 2(a)-(d). Noise that was caused by an inaccurate estimation of the centroid position was removed by applying a Savitzky-Golay filter. For the selected experimental runs, the settling velocities were averaged over the number of frames. Then the settling velocities obtained for individual particles were averaged over the number of experiments performed for each particle type.

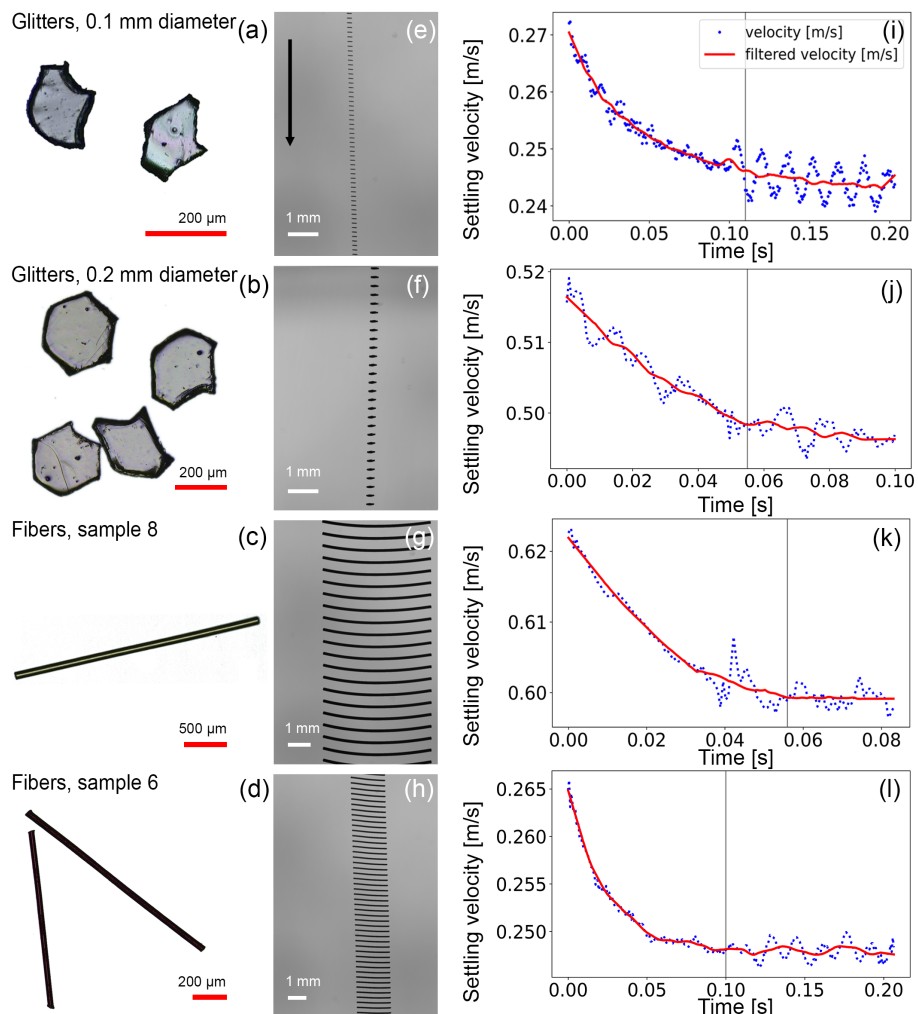

**Figure 2.** Example of the glitters and fibers and their settling behavior. (a)-(d): Microscopic images of glitter particles and fibers. (e)-(h): Overlays of particle images recorded by the cameras. The vertical vector shows the direction of gravity. (i)-(l): The postprocessed velocity time series of the particles. The dots represent the experimental postprocessed data, and the filtered data are shown as solid red lines. The right half of the dashed grey line in panels (i)-(l) indicates the data selected for the settling velocity analysis.

## 2.5 Drag coefficient of nonspherical particles

For describing the drag coefficient of spheres, accurate analytical and empirical models exist (Clift and Gauvin, 1971). For nonspherical particles, such generalized models do not exist; however, some empirical relationships have been determined as a function of particle shape. According to Coyle et al. (2023) and Saxby et al. (2018), the semi-empirical settling scheme of Bagheri and Bonadonna (2016, 2019) provides excellent results for the drag coefficient of nonspherical particles of various shapes. This scheme is suited for regular and irregular particle shapes settling in gas or liquids at Reynolds numbers smaller than

$3 \cdot 10^5$. The results are based on analytical solutions for ellipsoids in the Stokes' regime (Oberbeck, 1876) and measurements of the drag coefficient of 300 regular and irregular particles in air in settling columns and in a wind tunnel. Additionally, 881 experimental data points compiled from literature for particles of regular shapes in gases and liquids were considered. Bagheri and Bonadonna (2016, 2019)'s scheme is based on the Stokes $k_S$ and Newton drag corrections $k_N$, which represent the ratio of the drag coefficient of a nonspherical particle and the drag coefficient of a volume equivalent sphere in Stokes' and Newton's regime, respectively. These parameters are derived from Ganser (1993). Bagheri and Bonadonna (2016, 2019) introduced shape descriptors such as Stokes $F_S$ and Newton form factors $F_N$, which are based on the particle's volume-equivalent diameter $d_{eq}$, flatness ($f = S/I$), and elongation ($e = I/L$), where $L$, $I$, and $S$ are the particle's longest, intermediate, and shortest dimensions, respectively. These shape descriptors are easier to measure and correlate better with the Stokes $k_S$ and Newton drag corrections $k_N$ than sphericity, a widely used shape descriptor (Bagheri and Bonadonna, 2016). The chain of equations to calculate the drag coefficient can be found in Supplemental Materials (Eq. S1-S11). The model distinguishes between random-orientation-drag (i.e., drag coefficient $C_D$ of the particle when averaged over many random orientations), maximum-orientation-drag (i.e., drag coefficient when the particle's maximum projection area is normal to gravity), and also minimum-orientation-drag (i.e., drag coefficient when the particle's minimum projection area is normal to gravity). Additionally, an average between the predictions for random-orientation-drag and maximum-orientation-drag has been introduced by Tatsii et al. (2024), hereafter referred to as the average-orientation-drag model. There is also a simplified version of the model in which the shape of the particle is approximated by an ellipsoid of similar form by neglecting the term $\frac{d_{eq}^3}{L\,I\,S}$ in the Stokes form factor $F_S$ and in Newton form factor $F_N$ (Bagheri and Bonadonna, 2016, 2019). In this study, the full model version is used for both glitters and fibers, however, we also provide results using the simplified model for fibers, as suggested by Tatsii et al. (2024). The settling velocities of the volume-equivalent spheres are not measured but modeled, since the settling velocities of spheres are well known and need no further investigation.

## 2.6 Atmospheric transport modeling

We simulated the atmospheric transport of the glitter sizes used in the experiments, and also for the smallest commercially available glitter particles (0.05 mm nominal diameter; no experiments performed) using the Lagrangian transport and dispersion model FLEXPART (FLEXible PARTicle) version 11 (Bakels et al., 2024). The atmospheric transport of fibers has been investigated in detail before, for example by Tatsii et al. (2024). FLEXPART simulates the transport, turbulent diffusion, dry and wet deposition, decay, and first-order chemical reactions of tracers from local to global scales released from point, line, area, or volume sources (Pisso et al., 2019). Particles in FLEXPART are advanced by motion vectors combining the grid-scale wind velocity from linearly interpolated meteorological input data, the parameterized stochastic turbulent wind velocity, and, for aerosols, the settling velocity (Bakels et al., 2024). For calculating the setting velocities, the full, maximum-orientation-drag model of Bagheri and Bonadonna (2016, 2019) was used for these simulations.

Separate instantaneous releases of 10 000 particles were done twice per day at 03:00 and 15:00 local time on the 1st and 15th of each month for a one-year period. The particles were released from a vertical line source between 10 and 100 m above ground level from six different locations representing a range of meteorological conditions controlling wet and dry deposition:

London (51°30'N 0°7'W), Shanghai (31°13'N 121°28'E), Brasília (15°47'S 47°52'W), Cairo (30°1'N 31°14'W), New Orleans (29°57'N 90°4' W), and Reykjavík (64°7'N 21°49'W). The simulation times (listed in Table S2 in the supplementary materials) were selected based on preliminary test simulations to ensure that all airborne microplastics were deposited within the given time frames, while keeping computational cost at minimum. As input for the simulations, the most recent reanalysis dataset of the European Centre for Medium-Range Weather Forecasts, ERA5 (Hersbach et al., 2020), with hourly 0.5 ° x 0.5 ° horizontal resolution was used. The output was produced with 0.5 ° x 0.5 ° horizontal resolution and concentrations are sample from 6 seconds to 900 s, depending on particle size. Around the release points, nested grids were used with resolutions of 0.005, 0.0005, and 0.0001 ° for glitters and spheres with nominal diameters of 0.05 mm, 0.1 mm, and 0.2 - 3 mm, respectively.

Based on the simulations' output, average atmospheric residence times and transport distances of the particles were determined. The relative decrease in total atmospheric particle mass as a function of time $t$ was averaged over the number of releases and fitted to $y = e^{-\frac{t}{t_e}}$. Residence times were then determined as e-folding times $t_e$. The mean atmospheric transport distance $\bar{D}$ is calculated from the distance of grid cell $ij$ from the release point, $D_{ij}$, the total mass deposited in grid cell $ij$, $M_{ij}$, and the total mass deposited in the deposition field, $\sum_{ij} M_{ij}$ :

$$\bar{D} = \frac{\sum_{ij} D_{ij} \cdot M_{ij}}{\sum_{ij} M_{ij}}. \tag{2}$$

The deposition field represents the sum of dry and wet deposition fields. In FLEXPART, the dry deposition is treated as an exponential decay law of the dry deposition velocity, which is simulated with the resistance methodology, accounting for the aerodynamic and quasilaminar sublayer resistances, as well as the settling velocity (Slinn, 1982). The wet deposition scheme in FLEXPART distinguishes between in-cloud and below-cloud scavenging. The in-cloud scavenging coefficient depends on the scavenging ratio between the concentration of a substance in precipitation and the concentration in air, further outlined in Grythe et al. (2017), the precipitation rate, and the cloud depth where precipitation occurs (Bakels et al., 2024). Below-cloud scavenging depends on the relationship between aerosol and hydrometeor size and type, which is taken into account by the scheme of Wang et al. (2014) in FLEXPART. The removal of particle mass due to wet deposition is determined by an exponential decay function with the scavenging coefficient as decay constant. In the wet scavenging scheme of FLEXPART, the cloud condensation nuclei (CCN) and ice-nucleating particle (INP) efficiencies were set to 0.001 and 0.01, representing hydrophobic particles (Evangeliou et al., 2020). Scavenging efficiencies for rain and snow were assumed as 1, following Evangeliou et al. (2020). However, the exact scavenging efficiencies of microplastics are currently unknown. To address this uncertainty, we explored the sensitivity of FLEXPART to high CCN (0.5) and IN efficiencies (0.8) (Evangeliou et al., 2020). We also tested low scavenging efficiencies for rain (0.6) and snow (0.5) (Wang et al., 2014). We found that atmospheric lifetime and transport distances were relatively insensitive to the tested range of wet scavenging parameters, with a total variation below 5 % (Table S1). This is caused by the dominance of dry deposition driven by gravitational settling.

## 3 Results and discussion

### 3.1 Gravitational settling of glitters

The measured gravitational settling velocities for glitter particles range from $0.22 \, \mathrm{m\,s^{-1}}$ to $1.14 \, \mathrm{m\,s^{-1}}$, whereas the corresponding calculated settling velocities of volume-equivalent spheres range from $0.39 \, \mathrm{m\,s^{-1}}$ to $4.37 \, \mathrm{m\,s^{-1}}$ (Fig. 3a).

The gravitational settling velocities of glitters are reduced by 42 % to 74 % compared to volume-equivalent spheres (Table 1). The relative differences between the settling velocities of spherical and nonspherical particles tend to increase with increasing particle size and decreasing sphericity $\Psi$. The greatest reductions in settling velocity occur for glitters with the largest aspect

ratios ($L/S \geq 22$), where settling velocities are only 43 and 26 % of those of volume-equivalent spheres.

Figure 3(c) compares the observed gravitational settling velocities of the glitters with those predicted by the model of Bagheri and Bonadonna (2016, 2019), shown in Table 1. With average-orientation-drag configuration, the model of Bagheri and Bonadonna (2016, 2019) agrees well with the measurements, with a relative difference of 14.4 %. The highest deviations are received with random-orientation-drag configuration (19.0 %). This configuration performs well for large glitter particles

(nominal diameters > 0.4 mm), with a relative difference of 3.6 %, but performs less accurately for smaller particles (0.1 and 0.2 mm nominal diameters), with an average relative difference of 26.3 %. The best overall agreement is achieved for maximum-orientation-drag configuration, which yields an average relative difference of 13.2 %. This configuration shows the most accurate predictions for particles $\leq 0.2$ mm, with an average relative difference of 5.2 %. In general, the model of Bagheri and Bonadonna (2016, 2019) overestimates the settling velocities of small glitters and underestimates those of large glitters.

It should also be noted that the estimation of settling velocities is based on the mean of the measured dimensions and the nominal shapes of the particles given by the manufacturer, which may be subject to some variation, as shown in Fig. 2 and Fig. S1-S3. As a result, the discrepancies between the model of Bagheri and Bonadonna (2016, 2019) and the glitter particles are partly attributable to imperfections in the shape and size of the particles. It is important to note that the experimental results presented here are specific to still-air conditions. To understand how turbulence may alter settling behavior, Tinklenberg et al.

(2024) investigated the effect of turbulence on PET-disks between 0.3 and 3 mm falling in air. The smaller disks (0.3 and 0.5 mm nominal diameters) settled with the largest projection-area normal to the falling direction, independently of turbulence. The settling velocities of larger particles (nominal diameters of 1-3 mm) decreased in turbulent conditions, for the 3 mm disks influenced most significantly (up to 35 % compared to still-air conditions), which is attributed to drag nonlinearity. The authors report a much more randomized orientation distribution for millimetre-sized disks falling in turbulent air compared to quiescent

air. Rotation rates, however, were not significantly altered.

Overall, as particle size increases, the comparison of observed and modeled settling velocities suggests a shift from maximum-orientation-drag to average-orientation-drag, and then to random-orientation-drag aligning most closely. This can best be seen in Fig. 3(c), which compares modeled and observed settling velocities. In fact, in our experiments, we observed that the smallest glitters (0.1 and 0.2 mm nominal diameters) reached a terminal, steady-state orientation with their largest projection area perpendicular to the settling direction. Similar behavior has been observed by Bhowmick et al. (2024a, b) and Tinklenberg

et al. (2023) in air and by Goral et al. (2023) for flat disks, square plates, and irregularly shaped microplastics in distilled

**Table 1.** Characteristics of the Glitters and Their Modeled and Measured Settling Velocities[a]

| Nominal diameter | 0.05 mm | 0.1 mm | 0.2 mm | 0.4 mm | 0.6 mm | 1 mm | 3 mm |
|---|---|---|---|---|---|---|---|
| Symbol |  |  |  |  |  |  |  |
| $d_{eq}$ (μm) | 42.6 | 110.8 | 187.5 | 261.7 | 316.4 | 478.9 | 929.0 |
| $L$ (μm) | 50.0 | 175.8 | 305.9 | 496.5 | 672.2 | 1240.7 | 3287.1 |
| $I$ (μm) | 43.3 | 132.3 | 247.5 | 405.4 | 538.9 | 996.4 | 2683.1 |
| $S$ (μm) | 25.0 | 26.7 | 56.8 | 58.6 | 56.5 | 57.5 | 59.8 |
| $L/S$ | 2 | 7 | 5 | 8 | 12 | 22 | 55 |
| $fl$ | 0.58 | 0.20 | 0.23 | 0.14 | 0.10 | 0.06 | 0.02 |
| $el$ | 0.87 | 0.75 | 0.81 | 0.82 | 0.80 | 0.78 | 0.82 |
| $\Psi$ | 0.83 | 0.71 | 0.64 | 0.53 | 0.45 | 0.31 | 0.19 |
| $Re$ | 0.18 | 1.60 | 5.91 | 11.85 | 17.41 | 32.60 | 69.19 |
| $N$ | - | 11 | 5 | 3 | 3 | 2 | 3 |
| $v_t$ (m/s) | - | 0.221 | 0.482 | 0.663 | 0.842 | 1.041 | 1.139 |
| $\sigma_{vt}$ (m/s) | - | 0.014 | 0.048 | 0.004 | 0.019 | 0.114 | 0.007 |
| $v_{max}$ (m/s) | 0.067 | 0.243 | 0.488 | 0.608 | 0.654 | 0.779 | 0.927 |
| $v_{aver}$ (m/s) | 0.069 | 0.270 | 0.548 | 0.688 | 0.741 | 0.876 | 1.005 |
| $v_{rand}$ (m/s) | 0.072 | 0.308 | 0.637 | 0.812 | 0.875 | 1.019 | 1.107 |
| Symbol |  |  |  |  |  |  |  |
| $v_{sph}$ (m/s) | 0.073 | 0.393 | 0.837 | 1.271 | 1.564 | 2.404 | 4.373 |
| $v_t/v_{sph}$ | - | 0.56 | 0.58 | 0.52 | 0.54 | 0.43 | 0.26 |

[a] $d_{eq}$ is the volume equivalent diameter of a sphere, $L$ is the longest, $I$ the intermediate, and $S$ the shortest dimension of the particle. $fl$ is the particle's flatness and $el$ is its elongation. $\Psi$ is the sphericity, which compares the surface area of a particle to that of sphere $\pi d_{eq}^2/SA$, with $SA$ being the surface area of the particle. The Reynolds number $Re$ is defined as $\rho_a v_t d_{eq}/\nu$, where $\rho_a$ is the density and $\nu$ is the kinematic viscosity of the air. $N$ is the number of successful experiments. The measured settling velocity $v_t$ is given as the average over the number of experiments of the corresponding shape and size, with $\sigma_{vt}$ being the standard deviation. $v_{rand}$, $v_{max}$, $v_{aver}$, and $v_{sph}$ are the modeled velocities with random-, maximum-, and average-orientation-drag, and for volume equivalent spheres, respectively. For glitters with nominal diameters of 0.05 mm, no experiments but simulations have been performed, and their properties are listed for completeness sake.

water. In contrast, for larger glitter particles, tumbling motions occurred that increased with increasing particle size. This is in agreement with the observed shift of the best-suited particle-orientation model with particle size. In literature, a Reynolds number $Re \approx 100$ is quoted as a threshold below which particles tend to fall with their maximum projection area normal to the direction of gravity (Bagheri and Bonadonna, 2016; Willmarth et al., 1964). However, in our experiments, tumbling motions occur already at $Re < 100$, from about $Re = 17$ and becoming quite pronounced at $Re = 69$. Further investigation is needed to ascertain whether this discrepancy arises from differences in particle shapes or from variations in the experimental setups, each with its distinct capabilities and limitations.

## 3.2 Gravitational settling of fibers

The measured gravitational settling velocities for fibers range from $0.03 \text{ m s}^{-1}$ to $0.62 \text{ m s}^{-1}$. The corresponding calculated settling velocities of volume-equivalent spheres range from $0.10 \text{ m s}^{-1}$ to $1.93 \text{ m s}^{-1}$ (Fig. 3b). The settling velocities of fibers are reduced by 58 - 78 % compared to volume-equivalent spheres (Table 2). As with the glitters and in agreement with the results of Tatsii et al. (2024), the largest differences are observed for fibers with the largest aspect ratios ($L/S = 63, 121$).

Figure 3(d) presents a comparison of the modeled and experimental settling velocities of the fibers. The scheme of Bagheri and Bonadonna (2016, 2019) agrees best with the experiments with maximum-orientation-drag configuration, with a deviation of 12.4 %. For average-orientation-drag, the relative difference is 21.6 % and the largest deviation occurs again with random-orientation-drag (31.7 %).

In our experiments, most fibers achieved a terminal, steady-state orientation with their largest projected area normal to gravity, which is in agreement with previous work (Tatsii et al., 2024; Bhowmick et al., 2024a, b; Newsom and Bruce, 1994; Goral et al., 2023). This explains why the maximum-orientation-drag configuration works best for all particle sizes. The simplified average-orientation-drag model works well for fibers too (Table S3), similar to the study of Tatsii et al. (2024). For the simplified model, the best agreement is again achieved for maximum-orientation-drag configuration, with a relative difference of 7.3 %, followed by average-orientation-drag configuration (13.2 %). The largest difference occurs for random-orientation drag (24.3 %). However, Tatsii et al. (2024) investigated custom-produced fibers with sub-micron precision, minimizing the occurrence of imperfection and irregularities. In contrast, the commercially available fibers examined here are associated with certain deviations from the nominal size and shape, which introduces some uncertainties in the comparisons. Broadening the context to include turbulent conditions, Tatsii et al. (2024) compared the smallest atmospheric vortices (Kolmogorov microscales) with the properties of the microplastic fibers used in their experiments. Their conclusions are relevant for our fibers as well, as they are similar in size. They found that the time scales for alignment in fiber orientation are smaller than the smallest time scales typically encountered in atmospheric turbulence and that their settling velocities are larger than the Kolmogorov velocity scale. Therefore, they conclude that their measured settling velocities are representative of fibers in a turbulent atmosphere.

## 3.3 Impact on Atmospheric Transport

The good agreement between measured and modeled settling velocities allows for the use of the full Bagheri and Bonadonna (2016, 2019) scheme with maximum-orientation-drag for simulating the gravitational settling of glitters in the atmosphere. For this, the scheme has been implemented into the Lagrangian transport model FLEXPART. Figure 4 shows the atmospheric transport potential of glitters, represented by the annual mean of atmospheric horizontal travel distance and residence time, averaged over all six release points. Figures S6-S7 depict annual mean atmospheric travel distances and residence times for each release point separately. Differences in travel distance and residence time that occur between release points can be attributed to different meteorological conditions, causing differences, e.g., in vertical transport and, especially, wet deposition. In the following, we concentrate on the results averaged over all release sites.

**Table 2.** Characteristics of the Fibers and Their Modeled and Measured Settling Velocities[b]

| Name | Sample 1 | Sample 2 | Sample 3 | Sample 4 | Sample 5 | Sample 6 | Sample 7 | Sample 8 |
|---|---|---|---|---|---|---|---|---|
| Symbol | | | | | | | | |
| $d_{eq}$ (μm) | 56.5 | 107.6 | 108.4 | 117.9 | 175.5 | 193.1 | 280.4 | 435.7 |
| $L$ (μm) | 1200 | 2300 | 1500 | 1500 | 1500 | 2000 | 3000 | 5000 |
| $I = S$ (μm) | 10 | 19 | 24 | 27 | 49 | 49 | 70 | 105 |
| $L/S$ | 120 | 121 | 63 | 56 | 31 | 41 | 43 | 48 |
| $fl$ | 1 | 1 | 1 | 1 | 1 | 1 | 1 | 1 |
| $el$ | 0.01 | 0.01 | 0.07 | 0.02 | 0.03 | 0.02 | 0.02 | 0.02 |
| $\Psi$ | 0.11 | 0.06 | 0.05 | 0.05 | 0.03 | 0.03 | 0.02 | 0.01 |
| $Re$ | 0.11 | 0.49 | 0.50 | 0.85 | 2.87 | 3.16 | 6.97 | 17.66 |
| $N$ | 2 | 8 | 11 | 8 | 13 | 10 | 8 | 7 |
| $v_t$ (m/s) | 0.033 | 0.071 | 0.073 | 0.112 | 0.246 | 0.248 | 0.382 | 0.619 |
| $\sigma_{vt}$ (m/s) | 0.004 | 0.003 | 0.006 | 0.007 | 0.017 | 0.015 | 0.020 | 0.082 |
| $v_{max}$ (m/s) | 0.024 | 0.076 | 0.098 | 0.118 | 0.264 | 0.275 | 0.439 | 0.702 |
| $v_{aver}$ (m/s) | 0.026 | 0.085 | 0.110 | 0.132 | 0.298 | 0.311 | 0.498 | 0.800 |
| $v_{rand}$ (m/s) | 0.030 | 0.097 | 0.125 | 0.151 | 0.347 | 0.362 | 0.587 | 0.952 |
| Symbol | | | | | | | | |
| $v_{sph}$ (m/s) | 0.102 | 0.316 | 0.320 | 0.366 | 0.659 | 0.586 | 1.119 | 1.926 |
| $v_t/v_{sph}$ | 0.32 | 0.22 | 0.23 | 0.31 | 0.37 | 0.42 | 0.34 | 0.32 |

[b] $d_{eq}$ is the volume equivalent diameter of a sphere, $L$ is the longest, $I$ the intermediate, and $S$ the shortest dimension of the particle. $fl$ is the particle's flatness and $el$ is its elongation. $\Psi$ is the sphericity, which compares the surface area of a particle to that of sphere $\pi d_{eq}^2 / SA$, with $SA$ being the surface area of the particle. The Reynolds number $Re$ is defined as $\rho_a v_t d_{eq}/\nu$, where $\rho_a$ is the density and $\nu$ is the kinematic viscosity of the air. $N$ is the number of successful experiments. The measured settling velocity $v_t$ is given as the average over the number of experiments of the corresponding shape and size, with $\sigma_{vt}$ being the standard deviation. $v_{rand}$, $v_{max}$, $v_{aver}$, and $v_{sph}$ are the modeled velocities with random-, maximum-, and average-orientation-drag, and for volume equivalent spheres, respectively.

With increasing equivalent diameter, the particles' mean horizontal atmospheric travel distances resulting from the FLEX-PART simulations decrease (Fig. 4(a)): glitters with nominal diameters of 0.05, 0.1, 0.2, 0.4, 0.6, 1 and 3 mm travel 33.24, 1.52, 0.57, 0.48, 0.45, 0.38, and 0.21 km on average, and their volume-equivalent spheres travel 29.47, 1.04, 0.34, 0.22, 0.18, 0.11, 0.06 km, respectively. The smallest commercially available glitters therefore have the potential to pollute the areas surrounding the cities they are released in, while glitters larger than that are deposited close to their release points and will likely only affect the cities directly. However, resuspension of deposited particles, which is not considered in our simulations, may extend the effective transport distances substantially for all of the studied particles. E-folding times of glitters with nominal diameters of 0.05, 0.1, 0.2, 0.4, 0.6, 1 and 3 mm are 146.83, 12.25, 3.36, 2.78, 2.50, 1.67, 1.18 min, respectively, while the residence times of respective volume-equivalent spheres are 127.00, 6.42, 1.48, 0.77, 0.55, 0.28, 0.15 min.

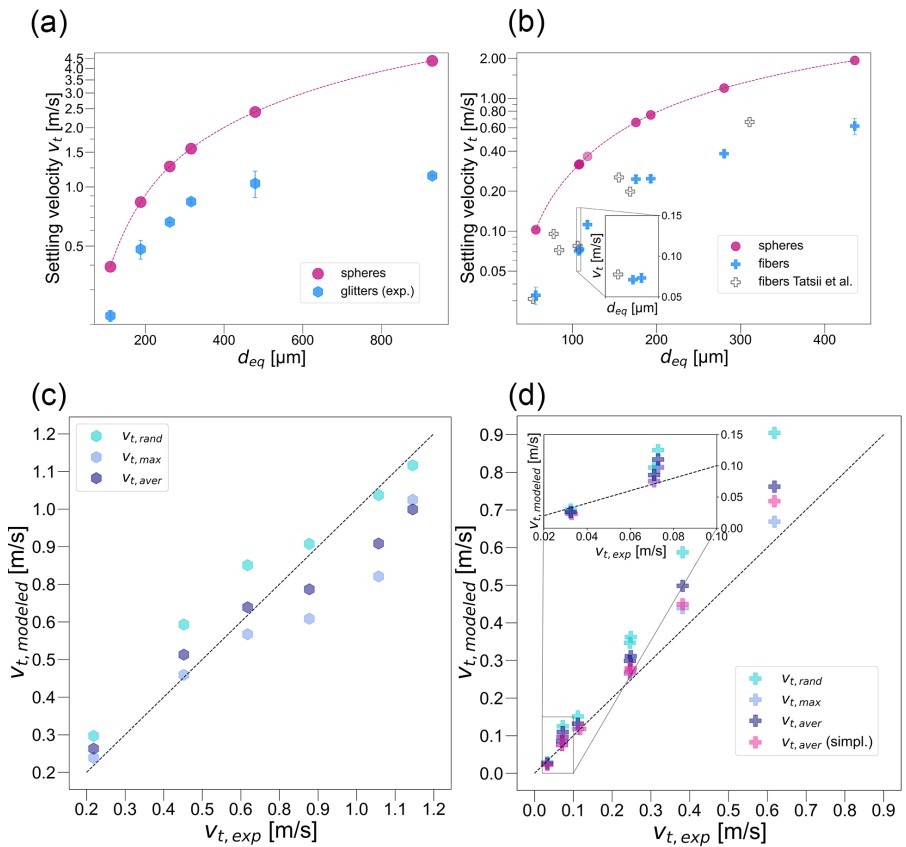

**Figure 3.** Observed and modeled settling velocities. Measured (blue symbols) settling velocities of glitters (a) and fibers (b) as a function of particle size, expressed as the diameter of a sphere of equivalent volume. Error bars depict the standard deviation. Modeled values of spheres of equivalent volume are shown with pink circles. Blue hexagons (a) and crosses (b) represent experimental results for glitters and fibers, respectively. White crosses in (b) depict previously published experimental results of settling velocities of straight fibers (Tatsii et al., 2024). The inset in (b) shows a closer view of the experimental data on fiber settling velocities. Scatter plot comparing the modeled and measured settling velocities of glitters (c) and fibers (d). Modeled values were calculated with the full model version for random (light blue), maximum-drag (blue) and averaged (dark blue) orientations. In (d), settling velocities of straight fibers modeled with the simplified, average-orientation-drag model (Tatsii et al., 2024) are depicted as well (pink crosses). The inset in (d) shows a closer view of the measured vs. modeled settling velocities of fibers. The dotted line shows the 1:1 relation.

Across all release points, the travel distances of 0.05, 0.1, 0.2, 0.4, 0.6, 1, and 3 mm glitters are $12 \pm 5$, $48 \pm 6$, $69 \pm 14$, $118 \pm 11$, $147 \pm 10$, $238 \pm 19$, and $261 \pm 48$ % greater than those of volume-equivalent spheres. This can be attributed to the longer atmospheric residence times of nonspherical particles in comparison to spherical ones (Fig. 4(b)). The mean atmospheric residence times of 0.05, 0.1, 0.2, 0.4, 0.6, 1, and 3 mm glitters are $7 \pm 14$, $90 \pm 9$, $128 \pm 11$, $264 \pm 41$, $346 \pm 49$, $477 \pm 99$, $699 \pm 213$ % longer compared to volume-equivalent spheres. Notice that, similar to the settling velocities, the

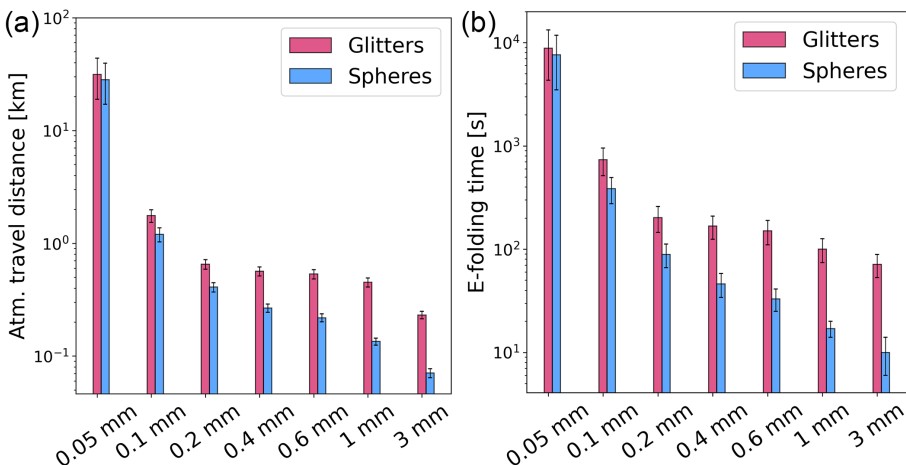

**Figure 4.** Atmospheric transport potential of glitters and volume-equivalent spheres. Shown are annual mean atmospheric horizontal travel distances (a) and atmospheric residence times (b) of glitters (pink) and spheres (blue). The results are averaged over all release points. Standard deviations are depicted by whiskers.

relative differences between spherical and nonspherical particles for both travel distances and residence times increase with decreasing sphericity.

We compare these results with findings from other modeling approaches: Saxby et al. (2018) find that nonspherical particles (sphericity = 0.5) travel 44 % further than spheres from their source. The authors used the atmospheric dispersion model NAME (Numerical Atmospheric-dispersion Modelling Environment (Jones et al., 2007)) and a measured shape parameter. Using Hybrid Single-Particle Lagrangian Integrated Trajectory model (HYSPLIT (Stein et al., 2015)) backward trajectory analyses, Wright et al. (2020) estimate that $100\,\mu\mathrm{m}$ spheres ($\rho = 1.05\,\mathrm{g/cm^3}$, $v_\mathrm{t} = 0.32\,m\,s^{-1}$) travel up to approximately 12 km. Important to note, however, that the settling velocities used by these authors have been estimated by Stokes' law. Long et al. (2022) utilized the Bagheri and Bonadonna (2016) scheme together with the Weather Research and Forecasting (WRF) model to predict the travel distances of microplastics. The fragments used by these authors show densities (1.1 - 1.2 $\mathrm{g/cm^3}$) and longest dimensions (60, 150, and 260 μm) comparable to some glitters in the current study. However, they have considerably smaller thickness (3 μm), sphericity (0.29, 0.18, and 0.12), and equivalent diameter (27, 51, and 73 μm) compared to glitters with similar longest dimensions. Therefore, for these fragments, the authors report settling velocities of 0.01 m/s and travel distances > 1000 km. A comparison to the results of Long et al. (2022) would therefore not make much sense. Instead, their results are included as a valuable demonstration of atmospheric transport potential of small fragments.

## 4 Conclusions

The gravitational settling velocities of glitter particles with nominal diameters between 0.1 mm and 3 mm and fibers with lengths between 1.2 and 5 mm were measured in a settling chamber. The observed settling velocities were compared to settling

velocities of volume-equivalent spheres and to modeled velocities using the Bagheri and Bonadonna (2016, 2019) shape-correction scheme. We have determined that glitters and fibers settle up to 74 % and 78 % slower compared to volume-equivalent spheres, respectively. The results are similar to those of Tatsii et al. (2024), who investigated the shape dependency of $C_D$ for 3D-printed, uniform microplastic fibers. They found that settling velocities of fibers are reduced by up to 76 % compared to volume-equivalent spheres, with highly elongated fibers exhibiting great atmospheric transport potential, capable of reaching remote locations such as the Arctic, when emitted in populated regions. The good agreement between modeled and observed settling velocities (average relative differences of 13.2 % for glitters and 12.4 % for fibers) shows that the model is suitable also for non-ideal, commercially available particles, which often have rough and complex surfaces. This is an important extension of the work of Tatsii et al. (2024) who only used microplastic fibers with perfectly defined shapes.

We use the Lagrangian transport model FLEXPART (Bakels et al., 2024) with an implemented shape correction scheme (Bagheri and Bonadonna, 2016, 2019) to simulate the atmospheric transport of glitter particles. By that, we have shown that commercial glitters have relatively large atmospheric transport potential, with the smallest particles traveling up to 33 km. Our simulation results also demonstrate that microplastic glitters are transported substantially farther than volume-equivalent spheres in the atmosphere: the atmospheric transport distances of microplastic glitters are up to 3.5 times greater than those of volume-equivalent spheres. This is a result of the longer atmospheric residence times of glitters, with e-folding times up to 7.4 times longer compared to spheres. These results indicate that glitter, when released into the air, may be transported on urban and regional scales. The same also holds true for microplastic fibers, as shown by Tatsii et al. (2024). This is particularly relevant for glitter released during parades or carnivals in cities, since airborne glitters may be deposited in soils and water bodies in and outside of the city, including also agricultural fields or conservation areas. Similarly, fibers released, for example from clothes, could easily be transported beyond the urban area. Especially for the smaller glitters and fibers, the atmosphere therefore constitutes an important transport medium, which is problematic since these particles are associated with the greatest harm for organisms and soils (Medyńska-Juraszek and Jadhav, 2022; Provenza et al., 2022). Furthermore, microplastic particles under weathering processes such as UV radiation, oxidation, or abrasion can degradate and be fragmented (Andrady, 2011), which will increase their transport potential. Resuspension from the ground would further enhance this, with particles potentially grasshopping several times before being incorporated into the soil matrix or a water body, which may - at least temporarily - halt the atmospheric transport.

Microplastic films or disks of higher aspect ratios and smaller sizes than considered here might even have larger atmospheric transport potential, constituting a possible field of further research. Overall, our results enhance the understanding of the shape-dependent settling behavior of airborne microplastics and can be used to constrain atmospheric transport models by accounting for such complex shapes as hexagonal disks and highly elongated fibers. The size and shape of the glitter particles used in this work are comparable to other aerosols, exhibiting disk-like shapes, such as plate crystals, mineral dust, pollen, or spores. This similarity extends the relevance of our work to these other types of particles.

*Data availability.* The data underlying the study is openly available on Zenodo and can be accessed via https://doi.org/10.5281/zenodo.15744633.

*Author contributions.* Conceptualization and research design: A.S., G.B., D.T., and A.R.; preparation of the experiments: A.R., T.B., and G.B.; experiments: A.R. and T.B.; postprocessing of the experimental data: G.B.; analysis of the experimental data: A.R., T.B., and G.B.; FLEXPART modelling: A.R., D.T., A.S.; analysis of the model results: A.R., D.T., A.S.; visualization: A.R; interpretation of the results: A.R., D.T., A.S., T.B., G.B; writing - original draft: A.R.; writing - review and editing: A.R., D.T., A.S., T.B., G.B.; supervision: D.T., A.S., T.B., and G.B.

*Competing interests.* The authors declare no competing financial interests.

*Acknowledgements.* The authors thank Alfredo Soldati, Giuseppe Carlo Alp Caridi, and Vlad Giurgiu from the Institute of Fluid Dynamics and Heat Transfer, TU Wien, for supplying the fibers and Alexander Bismarck and Qixiang Jiang from the Department of Materials Chemistry, University of Vienna, for providing the pycnometer for the density measurements of the glitter particles. This work was supported by the University of Vienna's research platform PLENTY - Plastics in the Environment and Society. We would like to thank the Max Planck Society and the MPI-DS for their support.

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
