# Peer review of "The atmospheric settling of commercially sold microplastics"

_EGUsphere, 2025_

## Author Comment (AC2)

**Responses to reviewer 1**

**Reviewer comments:**

1. *The current manuscript focused on a thorough investigation of the atmospheric settling behavior of microplastics, particularly glitters and fibers. The study is well-designed and well-written. And the authors have carried out extensive experimental work. In my view, the study would be highly valuable for publication. However, prior to a successful publication, I would recommend some changes to the manuscript.*

We sincerely thank the reviewer for taking the time to evaluate our manuscript thoroughly! The positive feedback and the constructive suggestions are very much appreciated. We carefully considered all comments and believe to have made the necessary revisions to address all comments.

Below we repeat the referee's comments in black italics, followed by our replies in blue regular font and quotes from the manuscript highlighted in green regular font.

2. *There are many abbreviations in the manuscript, please provide an abbreviation list at the begin of the paper.*

Thank you for the suggestion. We agree that too many abbreviations reduce the readability of the manuscript. In response to this comment, we reduced the number of abbreviations in the text to a minimum and made sure to define the remaining ones at first use, which we believe ensures readability without requiring such a dedicated list.

3. *A schematic diagram of the experimental setup is necessary in the '2.2 Experimental setup'.*

Thank you for pointing this out. We agree and added the schematic below (Figure 1 (b) in the revised manuscript) for clarification.

[Figure]

*Figure 1. Experimental Setup. (b) Schematic view of the experimental setup showing the two upper cameras (TX) and (TY), the two lower cameras (BX) and (BY), the mirror (M) arrangement, as well as the settling chamber (SC), the LED, and the illumination/imaging paths.*

4. *Please include a sensitivity analysis or error quantification in FLEXPART simulation.*

Thank you for this useful suggestion.
In response to this comment, we explored the sensitivity of the FLEXPART simulations to wet deposition, as the (currently unknown) exact scavenging efficiencies are a major uncertainty. We tested the sensitivity to low and high in-cloud and below-cloud scavenging efficiencies. The sensitivity analysis indicated a low impact of in-cloud and below-cloud scavenging for the coarse particles in this modeling setup.
The following statement has been added to the manuscript:

"In the wet scavenging scheme of FLEXPART, the cloud condensation nuclei (CCN) and ice-nucleating particle (INP) efficiencies were set to 0.001 and 0.01, representing hydrophobic particles. Scavenging efficiencies for rain and snow were assumed as 1. However, the exact scavenging efficiencies of microplastics are currently unknown. To address this uncertainty, we explored the sensitivity of FLEXPART to high CCN (0.5) and IN efficiencies (0.8) (Evangeliou et al., 2020). We also tested low scavenging efficiencies for rain (0.6) and snow (0.5) (Wang et al., 2014). We found that atmospheric lifetime and transport distances were relatively insensitive to the tested range of wet scavenging parameters, with a total variation below 5 % (Table S1). This is caused by the dominance of dry deposition driven by gravitational settling."

The following table has been added to the supplementary materials:

|  | CCN | IN | Crain | Csnow | Mean rel. difference travel distances (%) | Mean rel. difference residence times (%) |
|---|---|---|---|---|---|---|
| Base simulation | 0.001 | 0.01 | 1 | 1 | - | - |
| High CCN | 0.5 | 0.01 | 1 | 1 | 1.93 | 3.83 |
| High IN | 0.001 | 0.8 | 1 | 1 | 1.86 | 4.01 |
| Low. Crain | 0.001 | 0.01 | 0.6 | 1 | 1.71 | 3.82 |
| Low Csnow | 0.001 | 0.01 | 1 | 0.5 | 1.87 | 3.79 |

*Table S 1. Wet scavenging parameters that have been varied for the sensitivity test: cloud condensation nuclei efficiencies (CCN), ice-nucleating particle efficiencies (IN), and scavenging efficiencies for rain (Crain) and snow (Csnow). The parameters were selected based on the values proposed by Evangeliou et al. (2020) and Wang et al. (2014). Indicated are additionally the relative differences averaged over all particle sizes between the base simulations and sensitivity analyses.*

**References**

Evangeliou, N., Grythe, H., Klimont, Z., Heyes, C., Eckhardt, S., Lopez-Aparicio, S., & Stohl, A. (2020). Atmospheric transport is a major pathway of microplastics to remote regions. *Nature Communications*. doi:10.1038/s41467-020-17201-9

Wang, X., Zhang, L., & Moran, M. (2014). Development of a new semi-empirical parameterization for below-cloud scavenging of size-resolved aerosol particles by both rain and snow. *Geoscientific model development*. doi:10.5194/gmd-7-799-2014

---

## Author Comment (AC3)

**Responses to reviewer 3**

Reviewer comments:

1. *This work is an interesting and welcome contribution to investigating the dynamics of airborne microplastics. The scientific approach and the adopted methodologies are robust and sound, both on the experimental and modelling sides. At the same time, the impact of the paper would improve adding some more information and discussion, as detailed hereafter. This, to enable the reproducibility of the research and the comparison with the multiple studies that are emerging in relation to the atmospheric transport of microplastics.*

   *The manuscript can be considered and accepted for publication after (medium) revision.*

We sincerely thank the reviewer for taking the time to carefully read our manuscript and for the insightful feedback and recommendations! Thank you for the positive evaluation of our work while highlighting the fact that some more information and discussion are needed. We carefully considered all comments and believe to have made the necessary revisions to address all comments.

Below we repeat the referee's comments in black italics, followed by our replies in blue regular font and quotes from the manuscript highlighted in green regular font.

2. *L48-49: The number of publications related to the atmospheric dispersion of microplastics is rapidly increasing, and it is indeed difficult to keep continuously updated. However, some attention should be given to more recent works that might be of interest to this investigation, while here the first and by now 'classical' studies are recalled only.*

We thank the referee for pointing this out. We agree that our manuscript will benefit from including more recent literature and therefore added the following, more recent publications to our manuscript:

- Huang, Y., He, T., Yan, M., Yang, L., Gong, H., Wang, W., Qing, X., and Wang, J.: Atmospheric transport and deposition of microplastics in a subtropical urban environment, J. Hazard. Mater., 416, 126 168, https://doi.org/10.1007/s11869-024-01571-w, 2021.
- Mandal, M., Roy, A., Singh, P., and Sarkar, A.: Quantification and characterization of airborne microplastics and their possible hazards: a case study from an urban sprawl in eastern India, Front. Environ. Chem., Volume 5 - 2024, https://doi.org/10.3389/fenvc.2024.1499873,495, 2024.
- Martynova, A., Genchi, L., Laptenok, S. P., Cusack, M., Stenchikov, G. L., Liberale, C., and Duarte, C. M.: Atmospheric microfibrous deposition over the Eastern Red

Sea coast, Sci. Total Environ., 907, 167 902, https://doi.org/10.1016/j.scitotenv.2023.167902, 2024.

- Wright, S., Ulke, J., Font, A., Chan, K., and Kelly, F.: Atmospheric microplastic deposition in an urban environment and an evaluation of transport, Environ. Int., 136, 105 411, https://doi.org/10.1016/j.envint.2019.105411, 2020.
- Trainic, M., Flores, J. M., Pinkas, I., Pedrotti, M. L., Lombard, F., Bourdin, G., Gorsky, G., Boss, E., Rudich, Y., Vardi, A., and Koren, I.: Airborne microplastic particles detected in the remote marine atmosphere, Commun. Earth Environ 1, 64, https://doi.org/10.1038/s43247-020-00061-y, 2020.

3. *L81-84. Here the details of the experimental setup are kept at their minimum, then referring to two previous publications. Since an article should be as much as possible self-consistent, it would be worth adding some additional details as reported in those publications.*

We agree with the suggestions of the referee. Therefore, we added a schematic overview and a photo of the experimental setup (Figure 1) and the following description:

"The setup consists of an air-filled settling chamber, where the particles can settle under gravity, surrounded by four high-speed cameras. The settling chamber is a reinforced steel chamber with four glass windows, airtightly sealed edges, and dimensions of 90 · 90 · 200 mm in the X (direction of the light path from the LED), Y (horizontal direction orthogonal to X), and Z (direction of gravity) directions. A detachable bottom drawer allows for collection of the particles after an experiment. A particle injector is mounted on the settling chamber's removable top cover. The settling chamber is installed on a high-precision XYZ-stage, which allows movement of the settling chamber with 10 μm spatial resolution in all three directions (Bhowmick et al., 2024). A photograph of the settling chamber is shown in Fig. 1 (a).
The four high-speed cameras (Phantom VEO4K 990L, Vision Research) synchronized with a high-frequency pulsed white LED array (LED-Flashlight 300, model number 1103445, LaVision GmbH) surround the settling chamber, as shown in Fig. 1 (b). Only one camera receives direct light from the LED, the remaining three receive light reflected by mirrors. A waveform generator controls the pulse rate, amplitude, offset of the waveform, and duty cycle of the LED and creates a synchronization signal for the exposure times of all cameras. To control the cameras, the Phantom camera control (PCC) software was used. Further details of the setup, as well as the postprocessing and calibration processes, are described in Bhowmick et al. (2024) and Tatsii et al. (2024). For the experiments, the resolution of the cameras was set to 4096 x 1140 pixels, each pixel corresponding to a physical area of 6.75 x 6.75 μm$^2$, however, the smallest dimensions of the tested particles are larger than this value."

For clarity, we also extended the description of the experimental run:

"The settling chamber was positioned accordingly using the XYZ-stage to allow the cameras to capture particles near their steady-state velocity. After calibration, particles were introduced into the settling chamber using the particle injector. For this, a single particle was placed on top of a particle injector needle (a cylindrical rod with a length of 200 mm, a diameter of 12 mm, and a conical tip (Bhowmick et al., 2024)), which was then inserted into a needle guide. A release key that was placed into one of the needle grooves secured the needle in place. Upon removal, the needle dropped vertically until stopped by a needle block, causing the particle to detach. Different groves represent different particle initial velocities, ranging between 0.42 – 1.5 m/s (Bhowmick et al., 2024). Notice that insertion speeds are chosen to be close to expected terminal settling velocities, so particles can accelerate or decelerate in the air column, depending on whether insertion is slower or faster than the terminal settling velocity. As soon as the particle detached, cameras were activated by an external photoelectric trigger."

[Figure]

*Figure 1: Experimental setup. (a) Photo of the settling chamber (SC) with the particle injector and the XYZ-stage. Three cameras (TX, TY, and BY) and a mirror can be seen. (b) Schematic view of the experimental setup showing the two upper cameras (TX) and (TY), the two lower cameras (BX) and (BY), the mirror (M) arrangement, as well as the settling chamber (SC), the LED, and the illumination/imaging paths.*

4. *L110-120. A summary of the rationale, of the analytical and experimental approach behind Bagheri and Bonadonna 2016 model would be useful to better appreciate the improvement with respect to other (empirical) relationships.*

Thank you; we added the following description of rationale, experimental and analytical approach, describing the Bagheri and Bonadonna (2016, 2019) model further:

"This scheme is suited for regular and irregular particle shapes settling in gas or liquids at Reynolds numbers smaller than $3 \cdot 10^5$. The results are based on analytical solutions for ellipsoids in the Stokes' regime (Oberbeck, 1876) and measurements of the drag coefficient of 300 regular and irregular particles in air in settling columns and in a wind tunnel. Additionally, 881 experimental data points compiled from literature for particles of regular shapes in gases and liquids were considered. Bagheri and Bonadonna (2016, 2019)'s scheme is based on the Stokes $k_S$ and Newton drag corrections $k_N$, which represent the ratio of the drag coefficient of a nonspherical particle and the drag coefficient of a volume equivalent sphere in Stokes' and Newton's regime, respectively. These parameters are derived from Ganser (1993). Bagheri and Bonadonna (2016, 2019) introduced shape descriptors such as Stokes $F_S$ and Newton form factors $F_N$, which are based on the particle's volume-equivalent diameter $d_{eq}$, flatness ($f\ =\ S/I$), and elongation ($e\ =\ I/L$), where $L$, $I$, and $S$ are the particle's longest, intermediate, and shortest dimensions, respectively. These shape descriptors are easier to measure and correlate better with the Stokes $k_S$ and Newton drag corrections $k_N$ than sphericity, a widely used shape descriptor (Bagheri and Bonadonna, 2016). Equations describing the chain of equations to calculate the drag coefficient as a function of various shape descriptors introduced by Bagheri and Bonadonna (2016, 2019) can be found in Supplemental Materials (Eq. S1-S11)."

5. *L126-127. Since one of the main findings of this research is the much slower settling of glitter and fibres when compared to volume-equivalent spheres, it could be of interest for the authors to consider the results also from experimental studies treating fragments, characterised by irregular volume and shapes, for which the volume-equivalent concept may apply. In addition to the work by Preston et al. (2023) see, for instance, studies in a cylinder with water and in a wind tunnel:*

http://dx.doi.org/10.1016/j.envres.2023.115783 ;
https://doi.org/10.1016/j.hazadv.2024.100433

Thank you very much for this suggestion. In addition to Preston et al. (2023), we cited the work of Musso et al. (2024) as an example of studies performed on microplastics in air:

"Only a limited number of experimental studies have examined the settling behavior of microplastic films, disks, fibers, and fragments in air (Qi et al., 2012; Preston et al., 2023; Tatsii et al., 2024; Tinklenberg et al., 2024; Tinklenberg et al., 2023; Musso et al., 2024)."

We also considered the results of Goral et al. (2023). The following has been added:

"In fact, in our experiments, we observed that the smallest glitters (0.1 and 0.2 mm nominal diameters) reached a terminal, steady-state orientation with their largest projection area perpendicular to the settling direction. Similar behavior has been observed by Bhowmick et al. (2024), Bhowmick, Wang, Latt, & Bagheri (2024), Tinklenberg et al. (2023) in air and by Goral et al. (2023) for flat disks, square plates, and irregularly shaped microplastics in distilled water."

6. *L135-138. It is not clear (to me) how the simulations were performed. It is specified that the particles are released each 1^st and 15^th day of each month in the year 2020 at 03:00 and 15:00 local time: for how long? One time-step as a sort of puff release, or a continuous emission starting at that time and lasting 12 hours, or? How was the emission handled: as point releases centred on the city coordinates and spread along 10 to 100 m agl? As particles distributed in a volume emission, centred in the city area and in a layer between 10 and 100 m agl? More details are needed on the model configuration and runs, so that a reader can better understand and interpret the results of the model simulations.*

Thank you for pointing out that the model configuration is unclear. The following description of the model setup has been added in the text:

"Separate instantaneous releases of 10 000 particles were done twice per day at 03:00 and 15:00 local time on the 1st and 15th of each month for a one-year period. The particles were released from a vertical line source between 10 and 100 m above ground level from six different locations representing a range of meteorological conditions controlling wet and dry deposition: London (51°30'N 0°7'W), Shanghai (31°13'N 121°28'E), Brasília (15°47'S 47°52'W), Cairo (30°1'N 31°14'W), New Orleans (29°57'N 90°4' W), and Reykjavík (64°7'N 21°49'W). The simulation times (listed in Table S2 in the supplementary materials) were selected based on preliminary test simulations to ensure that all airborne microplastics were deposited within the given time frames, while keeping computational cost at minimum."

The following table has been added to the supplementary material:

| Particle | Simulation time |
|---|---|
| 0.05 mm diameter glitters | 3 days |
| 0.1 mm diameter glitters | 10 h |
| 0.2 mm diameter glitters | 10 h |
| 0.4 mm diameter glitters | 5 h |
| 0.6 mm diameter glitters | 3 h |
| 1 mm diameter glitters | 2 h |
| 3 mm diameter glitters | 2 h |
| 0.05 mm diameter spheres | 3 days |
| 0.1 mm diameter spheres | 10 h |
| 0.2 mm diameter spheres | 3 h |

| 0.4 mm diameter spheres | 2 h |
|---|---|
| 0.6 mm diameter spheres | 2 h |
| 1 mm diameter spheres | 2 h |
| 3 mm diameter spheres | 2 h |

*Table S 2: Simulation times for different FLEXPART runs.*

7. *L145-148. FLEXPART is a well-known and largely used model, so it can be accepted that no specific details are provided on the physics in the model. However, here the total mass deposited plays an important role, as in eq. (2), and briefly recalling how the dry and wet depositions are treated in the model would be useful.*

We thank the referee for their comment. In response to this comment, we added a description of the dry and wet deposition scheme and of the total mass deposited to the methodology:

"Based on the simulations' output, average atmospheric residence times and transport distances of the particles were determined. The relative decrease in total atmospheric particle mass as a function of time $t$ was averaged over the number of releases and fitted to $y = e^{-\frac{t}{t_e}}$. Residence times were then determined as e-folding times $t_e$. The mean atmospheric transport distance $\overline{D}$ is calculated from the distance of grid cell $ij$ from the release point, $D_{ij}$, the total mass deposited in grid cell $ij$, $M_{ij}$, and the total mass deposited in the deposition field, $\sum_{ij} M_{ij} : \overline{D} = \frac{\sum_{ij} D_{ij} \cdot M_{ij}}{\sum_{ij} M_{ij}}$.

The deposition field represents the sum of dry and wet deposition fields. In FLEXPART, the dry deposition is treated as an exponential decay law of the dry deposition velocity, which is simulated with the resistance methodology, accounting for the aerodynamic and quasilaminar sublayer resistances, as well as the settling velocity (Slinn, 1982). The wet deposition scheme in FLEXPART distinguishes between in-cloud and below-cloud scavenging. The in-cloud scavenging coefficient depends on the scavenging ratio between the concentration of a substance in precipitation and the concentration in air, further outlined in Grythe et al. (2017), the precipitation rate, and the cloud depth where precipitation occurs (Bakels et al., 2024). Below-cloud scavenging depends on the relationship between aerosol and hydrometeor size and type, which is taken into account by the scheme of Wang et al. (2014) in FLEXPART. The removal of particle mass due to wet deposition is determined by an exponential decay function with the scavenging coefficient as decay constant."

8.  *L151-152 and 182-183. In general, and given all potential uncertainties, I wonder whether three decimals are truly significant/substantial to distinguish the measured and calculated settling velocities.*

Thank you for pointing this out. We agree and will reduce it to two decimals, as it increases readability as well.

9.  *L168-169 and 196-198. A partial interpretation of the discrepancies between the model and the measurements is attributed to the imperfections in the shape and size of the glitter particles or fibres.*
    *A discussion on the possible uncertainties related to the experimental approach, leading to uncertainties also in the observed data, would be worth it.*
    *The experiments are conducted in quiet air. It would be interesting to provide some comments on the potential/expected effect of turbulence on the settling dynamics.*

We thank the reviewer for their suggestion. We added the following paragraph to the Methods section:

**"Verification of the setup**

The sensitivity tests of the experimental setup by Bhowmick et al. (2024) showed that possible uncertainties in measurements caused by a change of temperature due to illumination and airflow caused by needle movement are negligible. Bhowmick et al. (2024) and Tatsii et al. (2024) both thoroughly verified the setup by dropping spheres and nonspherical particles of different diameters and comparing the measured settling velocities to the empirical model of Clift and Gauvin (1971) and to the direct numerical simulations of Bhowmick et al. (2025). The authors report relative errors below 5 %. We point out that the stated error of the Clift and Gauvin (1971) model is about 6 % at Reynold's numbers below $3 \cdot 10^5$ (Clift and Gauvin, 1971) and the average error of the Bagheri and Bonadonna (2016) model is 10 %."

To discuss the effect of turbulence, we added the following:

"It is important to note that the experimental results presented here are specific to still-air conditions. To understand how turbulence may alter settling behavior, Tinklenberg et al. (2024) investigated the effect of turbulence on PET-disks between 0.3 and 3 mm falling in air. The smaller disks (0.3 and 0.5 mm nominal diameters) settled with the largest projection-area normal to the falling direction, independently of turbulence. The settling velocities of larger particles (nominal diameters of 1-3 mm) decreased in turbulent conditions, for the 3 mm disks influenced most significantly (up to 35 % compared to still-air conditions), which is attributed to drag nonlinearity. The authors report a much more randomized orientation distribution for millimetre-sized disks falling in turbulent air compared to quiescent air. Rotation rates, however, were not significantly altered.

Broadening the context to include turbulent conditions, Tatsii et al. (2024) compared the smallest atmospheric vortices (Kolmogorov microscales) with the properties of the microplastic fibers used in their experiments. Their conclusions are relevant for our fibers as well, as they are similar in size. They found that the time scales for alignment in fiber orientation are smaller than the smallest time scales typically encountered in atmospheric turbulence and that their settling velocities are larger than the Kolmogorov velocity scale. Therefore, they conclude that their measured settling velocities are representative of fibers in a turbulent atmosphere."

10. *L239-247. Consider comparing the distances travelled by the particles and fibres in your study with findings from other studies in the literature. In this relatively recent field of study, it is important to verify similarities, convergences and differences, to evaluate whether the scientific community is following a sensible research path.*

Thank you for this useful remark. We expanded the discussion and added the following paragraph:

"We compare these results with findings from other modeling approaches: Saxby et al. (2018) find that nonspherical particles (sphericity = 0.5) travel 44 % further than spheres from their source. The authors used the atmospheric dispersion model NAME and a measured shape parameter.
Using Hybrid Single-Particle Lagrangian Integrated Trajectory model (HYSPLIT) backward trajectory analyses, Wright et al. (2020) estimate that 100 µm spheres ($\rho$ = 1.05 g/cm$^3$, $v_t$ = 0.32 m s$^{-1}$) travel up to approximately 12 km. Important to note, however, that the settling velocities used by these authors have been estimated by Stokes' law.
Long et al. (2022) utilized the Bagheri and Bonadonna (2016) scheme together with the Weather Research and Forecasting (WRF) model to predict the travel distances of microplastics. The fragments used by these authors show densities (1.1 - 1.2 g/cm$^3$) and longest dimensions (60, 150, and 260 µm) comparable to some glitters in the current study. However, they have considerably smaller thickness (3 µm), sphericity (0.29, 0.18, and 0.12), and equivalent diameter (27, 51, and 73 µm) compared to glitters with similar longest dimensions. Therefore, for these fragments, the authors report settling velocities of 0.01 m/s and travel distances > 1000 km. A comparison to the results of Long et al. (2022) would therefore not make much sense. Instead, their results are included as a valuable demonstration of atmospheric transport potential of small fragments."

11. *Supplementary material: please note that the citation of Bagheri and Bonadonna (2016, 2019) is repeated twice every time (a latex trap, I guess).*

Thank you very much for noticing. This has been changed in the revised manuscript.

**References**

Bagheri, G., & Bonadonna, C. (2016). On the drag of freely falling non-spherical particles. *Powder Technology, 301*, 526-544. doi:10.1016/j.powtec.2016.06.015

Bagheri, G., & Bonadonna, C. (2019). Erratum to "On the drag of freely falling non-spherical particles" [Powder Technology 301 (2016) 526–544, DOI: 10.1016/j.powtec.2016.06.015]. *Powder Technol., 349*, 108. doi:10.1016/j.powtec.2018.12.040

Bagheri, G., Bonadonna, C., Manzella, I., Pontelandolfo, P., & Haas, P. (2013). Dedicated vertical wind tunnel for the study of sedimentation. *Rev. Sci. Instrum., 84*(5), 054501. doi:10.1063/1.4805019

Bakels, L., Tatsii, D., Tipka, A., Thompson, R., Dütsch, M., Blaschek, M., . . . Stohl, A. (2024). FLEXPART version 11: Improved accuracy, efficiency, and flexibility. *EGUsphere*, 1-50. doi:10.5194/egusphere-2024-1713

Bhowmick, T., Latt, J., Wang, Y., & Bagheri, G. (2025). Palabos Turret: A particle-resolved numerical framework for settling dynamics of arbitrary-shaped particles. *Comput. Fluids*. doi:10.1016/j.compfluid.2025.106696

Bhowmick, T., Seesing, J., Gustavsson, K., Guettler, J., Wang, Y., Pumir, A., . . . Bagheri, G. (2024). Inertia Induces Strong Orientation Fluctuations of Nonspherical Atmospheric Particles,. *Phys. Rev. Lett., 132*, 034 101. doi:10.1103/PhysRevLett.132.034101

Bhowmick, T., Wang, Y., Latt, J., & Bagheri, G. (2024). Twist, turn and encounter: the trajectories of small atmospheric particles unravelled. doi:10.48550/arXiv.2408.11487

Clift, R., & Gauvin, W. (1971). otion of entrained particles in gas streams,. *CJCE, 49*, 439-448. doi:10.1002/cjce.5450490403

Ganser, G. (1993). A rational approach to drag prediction of spherical and nonspherical particles. *Powder Technol., 77*, 143-152. doi:10.1016/0032-5910(93)80051-B

Goral, K., Guler, H., Larsen, B., Carstensen, S., Christensen, E., Kerpen, N., . . . Fuhrmann, D. (2023). Settling velocity of microplastic particles having regular and irregular shapes. *Environ. Res.*, 115783. doi:10.1016/j.envres.2023.115783

Grythe, H., Kristiansen, N., Groot Zwaaftink, C., Eckhardt, S., Ström, J., Tunved, P., . . . Stohl, A. (2017). A new aerosol wet removal scheme for the Lagrangian particle model FLEXPART v10. *Geosci. Model Dev.*, 1447-1466.

Kowalski, N., Reichardt, A., & Waniek, J. (2016). Sinking rates of microplastics and potential implications of their alteration by physical, biological, and chemical factors. *Mar. Pollut. Bull., 109*, 310-319. doi:10.1016/j.marpolbul.2016.05.064

Long, X., Fu, T., Yang, X., Tang, Y., Zheng, Y., Zhu, L., . . . Li, B. (2022). Efficient Atmospheric Transport of Microplastics over Asia and Adjacent Oceans. *Environ. Sci. Technol.* doi:10.1021/acs.est.1c07825

Musso, M., Harms, F., Martina, M., Fischer, E., Leitl, B., & Trini Castelli, S. (2024). Experimental investigation of the fallout dynamics of microplastic fragments in wind tunnel: The BURNIA agenda. *J. Hazard. Mater. Adv., 14*, 100 433. doi:10.1016/j.hazadv.2024.100433

Oberbeck, A. (1876). Ueber stationäre Flüssigkeitsbewegungen mit Berücksichtigung der inneren Reibung. *Journal für reine und angewandte Mathematik, 81*, 62-80.

Preston, C., McKenna Neumann, C., & Aherne, J. (2023). Effects of Shape and Size on Microplastic Atmospheric Settling Velocity. *Environ. Sci. Technol.*, 11 937-11 947. doi:10.1021/acs.est.3c03671

Qi, G., Nathan, G., & Kelso, R. (2012). PTV measurement of drag coefficient of fibrous particles with large aspect ratio. *Powder Technol., 229*, 261-269. doi:10.1016/j.powtec.2012.06.049

Saxby, J., Beckett, F., Cashman, K., Rust, A., & Tennant, E. (2018). The impact of particle shape on fall velocity: Implications for volcanic ash dispersion modelling. *J. Volcanol. and Geotherm. Res.* doi:10.1016/j.jvolgeores.2018.08.006

Schneider, C., Rasband, W., & Eliceiri, K. (2012). NIH Image to ImageJ: 25 years of image analysis. *Nat. Methods*.

Slinn, W. (1982). Predictions for particle deposition to vegetative canopies. *Atmos. Environ., 16*, 1785-1794.

Tatsii, D., Bucci, S., Bhowmick, T., Guettler, J., Bakels, L., Bagheri, G., & Stohl, A. (2024). Shape Matters: Long-Range Transport of Microplastic Fibers in the Atmosphere. *Environ. Sci. Technol., 58*, 671-682. doi:10.1021/acs.est.3c08209

Tinklenberg, A., Guala, M., & Coletti, F. (2023). Thin disks falling in air. *J. Fluid. Mech., 962*. doi:10.1017/jfm.2023.209,

Tinklenberg, A., Guala, M., & Coletti, F. (2024). Turbulence effect on disk settling dynamics. *J. Fluid Mech*. doi:10.1017/jfm.2024

Wang, X., Zhang, L., & Moran, M. (2014). Development of a new semi-empirical parameterization for below-cloud scavenging of size-resolved aerosol particles by both rain and snow. *Geosci. Model Dev., 7*, 799-818. doi:10.5194/gmd-7-799-2014

Wang, Z., Dou, M., Ren, P., Sun, B., Jia, R., & Zhou, Y. (2021). Settling velocity of irregularly shaped microplastics under steady and dynamic flow conditions. *ESPR*.

Wright, S., Ulke, J., Font, A., Chan, K., & Kelly, F. (2020). Atmospheric microplastic deposition in an urban environment and an evaluation of transport. *Environ. Int., 136*, 105 411. doi:10.1016/j.envint.2019.105411